# Leptin receptor neurons in the dorsomedial hypothalamus regulate diurnal patterns of feeding, locomotion, and metabolism

Chelsea L Faber[1]*, Jennifer D Deem[1], Bao Anh Phan[1], Tammy P Doan[1], Kayoko Ogimoto[1], Zaman Mirzadeh[2], Michael W Schwartz[1], Gregory J Morton[1]*

[1]UW Medicine Diabetes Institute, Department of Medicine, University of Washington, Seattle, United States; [2]Department of Neurosurgery, Barrow Neurological Institute, Phoenix, United States

**Abstract** The brain plays an essential role in driving daily rhythms of behavior and metabolism in harmony with environmental light–dark cycles. Within the brain, the dorsomedial hypothalamic nucleus (DMH) has been implicated in the integrative circadian control of feeding and energy homeostasis, but the underlying cell types are unknown. Here, we identify a role for DMH leptin receptor-expressing (DMH$^{LepR}$) neurons in this integrative control. Using a viral approach, we show that silencing neurotransmission in DMH$^{LepR}$ neurons in adult mice not only increases body weight and adiposity but also phase-advances diurnal rhythms of feeding and metabolism into the light cycle and abolishes the normal increase in dark-cycle locomotor activity characteristic of nocturnal rodents. Finally, DMH$^{LepR}$-silenced mice fail to entrain to a restrictive change in food availability. Together, these findings identify DMH$^{LepR}$ neurons as critical determinants of the daily time of feeding and associated metabolic rhythms.

*For correspondence:
kasperc@uw.edu (CLF);
gjmorton@uw.edu (GJM)

**Competing interests:** The authors declare that no competing interests exist.

## Introduction

Synchrony between behavior and environmental rhythms enables animals to predict food availability and optimize metabolism in anticipation of daily periods of fasting and feeding (*Saper et al., 2005*). Conversely, mistimed feeding (i.e., food consumption during the normal resting period) impairs metabolism and increases susceptibility to obesity and associated metabolic impairment (*Greco and Sassone-Corsi, 2019*; *Stephan et al., 1979*). While the hypothalamic suprachiasmatic nucleus (SCN) is well known to entrain circadian rhythmicity in accordance with light–dark cycles, food availability can also entrain metabolic rhythms independently from the SCN (*Greco and Sassone-Corsi, 2019*). Illustrating this point, although rodents with SCN lesions exhibit profound disruptions in circadian rhythms, they retain the ability to retrain metabolic and behavioral rhythms in accordance with a scheduled meal (*Stephan et al., 1979*). Moreover, scheduled feeding has no effect on rhythmic gene expression in the SCN (*Damiola, 2000*), suggesting the existence of extra-SCN food-entrainable oscillators that function to align behavior and metabolism with food availability (*Saper et al., 2005*). Although somewhat controversial (*Landry et al., 2006*), evidence suggests the dorsomedial hypothalamic nucleus (DMH) may play such a role. First, the DMH receives both direct and indirect input from the SCN (*Watts et al., 1987*), and DMH neurons, in turn, project to neurons in brain areas regulating metabolism and feeding, including the arcuate nucleus (ARC; *Garfield et al., 2016*; *Gautron et al., 2010*). Moreover, DMH lesioning in rats not only disrupts circadian rhythms in feeding, locomotion, and core temperature (*Gooley et al., 2006*; *Chou et al., 2003*) but also precludes entrainment to scheduled feeding (*Chou et al., 2003*). However, the relevant DMH cell types

mediating these effects are unknown. Based on recent evidence that DMH neurons expressing leptin receptor (DMH[LepR]) are both sensitive to food availability and make inhibitory synaptic connections with agouti-related protein (AgRP) neurons to modulate feeding (*Garfield et al., 2016*), we identified DMH[LepR] neurons as a candidate population for circadian control of food intake and associated metabolic rhythms.

## Results and discussion

### Silencing DMH[LepR] neurons elicits transient hyperphagia and increased adiposity

To determine the role of DMH[LepR] neurons in feeding and metabolism, we used a viral loss-of-function approach (*Figure 1A*). Specifically, synaptic neurotransmission by DMH[LepR] neurons was permanently blocked by bilateral microinjection into the DMH of an adeno-associated virus (AAV) encoding Cre-dependent tetanus toxin light chain fused with a GFP reporter (AAV1-CBA-DIO-GFP: TeTx; *Kim et al., 2009*; *Han et al., 2015*; *Campos et al., 2017*). Viral transduction was confirmed by histochemical detection of GFP in the DMH (*Figure 1B–C*); as expected, GFP was undetected in Cre-negative controls (not shown). Although GFP+ cell bodies were not detected outside of the DMH, abundant GFP+ terminals were detected in the ARC (*Figure 1B–C*), consistent with previous evidence of an inhibitory DMH[LepR]→ ARC[AgRP] neurocircuit implicated in feeding control (*Garfield et al., 2016*; *Krashes et al., 2014*).

Whereas previous evidence showed no effect of acute inhibition of DMH[LepR] neurons on feeding (*Garfield et al., 2016*), chronic silencing of DMH[LepR] neurons resulted in hyperphagia that was sustained for several days (*Figure 1D–E*). This transient hyperphagic response was associated with sustained weight gain (*Figure 1F*) and a modest increase in adipose mass (*Figure 1G*) that persisted despite daily food intake eventually falling below that of controls (*Figure 1E*). Consistent with the increased adiposity, we also detected modestly increased plasma leptin levels (*Figure 1H*) and elevated fasting levels of both blood glucose (control vs. TeTx: $72.0 \pm 5.6$ vs. $107.1 \pm 7.3$, $t_{9.969}$=3.816; p=0.003) and plasma insulin (control vs. TeTx: $0.49 \pm 0.04$ vs. $1.24 \pm 0.12$, $t_{6.092}$=5.807; p=0.001), suggestive of insulin resistance. These findings extend and refine previous work implicating a physiological role for DMH[LepR] neurons in energy homeostasis (*Garfield et al., 2016*; *Rezai-Zadeh et al., 2014*).

### Neurotransmission by DMH[LepR] neurons is required for suppression of feeding by leptin

Previous work has shown both that leptin treatment depolarizes DMH neurons expressing LepR (*Simonds et al., 2014*) and that leptin receptor signal transduction in GABAergic DMH neurons is required for the acute anorectic effect of leptin (*Xu et al., 2018*). To investigate the possibility that leptin's anorectic effects involve activation of DMH[LepR] neurons, we next tested whether DMH[LepR] inactivation blunts leptin-mediated anorexia. First, the specificity of GFP:TeTx expression in DMH[LepR] neurons was confirmed by establishing that leptin-induced pSTAT3, a marker of LepR signal transduction, colocalizes with virally transduced cells following systemic leptin injection (*Figure 2A*). Next, control and DMH[LepR]-silenced mice were fasted for 24 hr, followed by i.p. injection of either leptin or saline vehicle, after which food was returned. Interestingly, control animals lost more weight during the fast (*Figure 2B*) and consequently exhibited a greater refeeding response following saline treatment than TeTx mice (*Figure 2C*; dashed bars). Nonetheless, the effect of leptin to further suppress food intake was readily detected in controls but absent in DMH[LepR]-silenced mice (*Figure 2C*; solid bars). These findings extend previous evidence (*Xu et al., 2018*) suggesting a key role for DMH[LepR] neurons in leptin-mediated suppression of fasting-induced refeeding.

We note that since DMH[LepR] neurons directly synapse onto and inhibit AgRP neurons (*Garfield et al., 2016*), TeTx-mediated DMH[LepR] silencing is predicted to increase AgRP neuron activity. As AgRP activation not only increases feeding (*Aponte et al., 2011*) but also promotes de novo lipogenesis and suppresses lipolysis during fasting (*Cavalcanti-de-Albuquerque et al., 2019*), AgRP disinhibition offers a feasible explanation for the observed effects of DMH[LepR] silencing not only to promote hyperphagia and weight gain, but also to preserve body weight during fasting. Future studies are warranted to test this possibility.

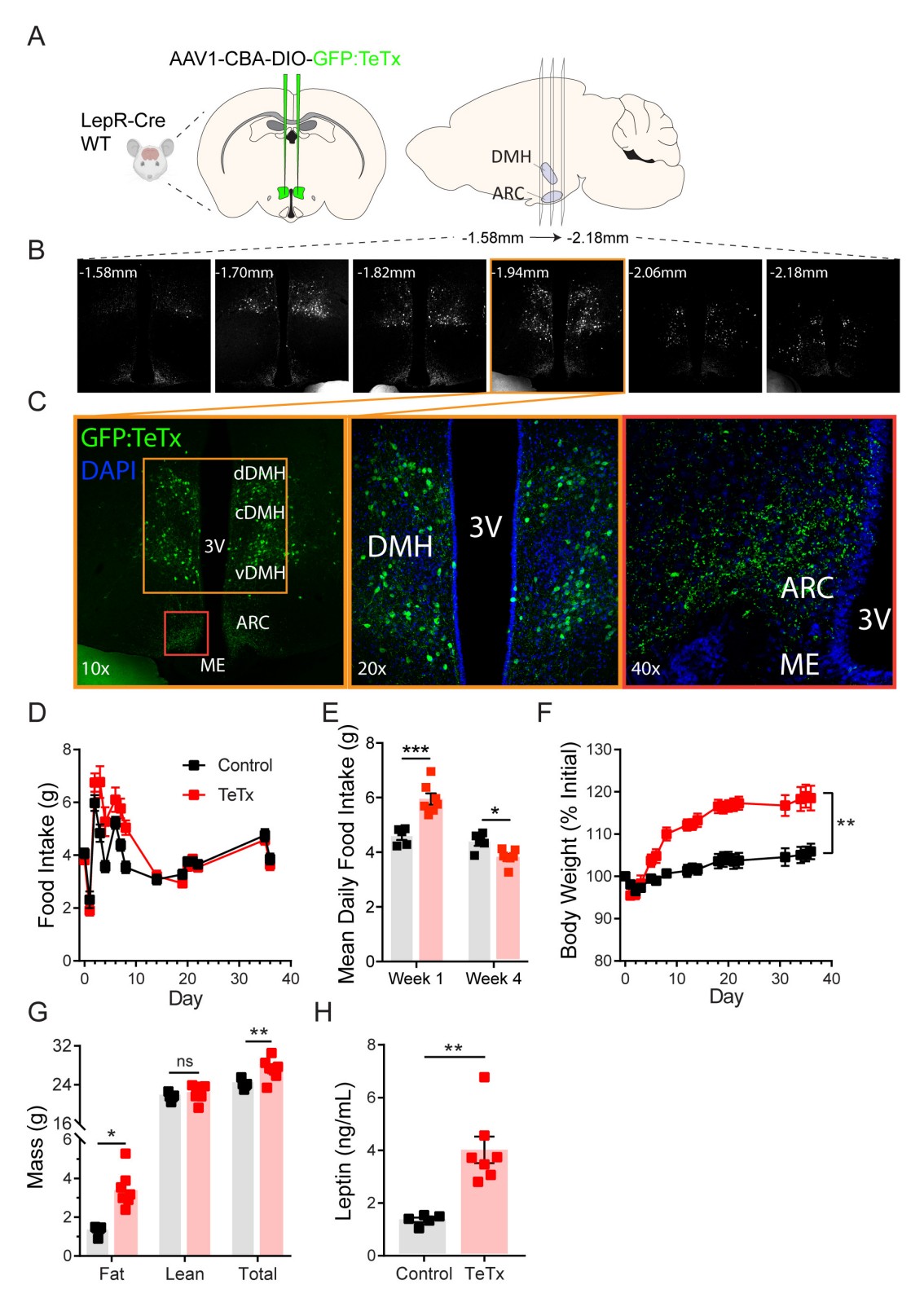

**Figure 1.** Silencing DMH^LepR neurons elicits transient hyperphagia and increased adiposity in adult male mice. (**A**) Experimental schematic for chronic inhibition of DMH^LepR by microinjection on day 0 of an AAV1 containing a Cre-dependent GFP-fused TeTx delivered bilaterally to the DMH of LepR-Cre+ male mice (TeTx; n=7) and Cre-negative littermate controls (control; n=5). (**B**) Stereological fluorescent images from a representative animal showing the rostral-caudal extent of GFP:TeTx expression. (**C**) Left: colorized, higher magnification view of the boxed orange region from (**B**). Middle:
*Figure 1 continued on next page*

*Figure 1 continued*

higher magnification view of the boxed orange region showing neuronal cell bodies targeted within the DMH. Right: higher magnification view of the boxed red region showing GFP:TeTx+ terminals of targeted DMH[LepR] neurons within the arcuate nucleus (ARC). (D) Mean daily food intake following viral microinjection. Two-way ANOVA: $F_{(1,10)}=4.658$; p=0.0563 (main effect of TeTx); $F_{(14,140)}=4.886$; p<0.0001 (time x TeTx interaction). (E) Mean daily food intake from week 1 relative to week 4. Two-way ANOVA: $F_{(1,10)}=5.575$; p=0.0399 (main effect of TeTx); $F_{(1,10)}=39$; p<0.001 (time x TeTx interaction). (F) Body weight expressed as %day 0 value. Two-way ANOVA: $F_{(1,10)}=20.18$; p=0.0012 (main effect of TeTx). $F_{(19,190)}=14.67$; p<0.0001 (time x TeTx interaction). (G) Fat, lean, and total mass 26 days after viral microinjection. Multiple t-tests; $t_{fat}=4.847$; p=0.0014; $t_{total}=2.884$; p=0.016. (H) Plasma leptin 21 days after viral microinjection. Unpaired t-test, t=5.17, p=0.0017. Data are mean ± SEM. For repeated measures, post hoc Sidak's test for each time point is indicated on the graph. *p<0.05,**p<0.01, ***p<0.001, ****p<0.0001.

The online version of this article includes the following source data and figure supplement(s) for figure 1:

**Source data 1.** Longitudinal measures following DMH-LepR TeTx.

**Figure supplement 1.** Representative viral expression is evident in both the ventral and dorsal compartments of the dorsomedial hypothalamic nucleus (DMH) following microinjection of GFP:TeTx to the DMH of LepR-Cre+ male mice.

**Figure supplement 2.** Silencing DMH[LepR] neurons in female mice recapitulates body weight and fat mass increase observed in males, but not acute hyperphagia.

## DMH[LepR] inactivation disrupts diurnal feeding, locomotion, and metabolic rhythms

To determine whether the observed impairments in energy homeostasis were associated with changes in diurnal rhythmicity in normal (14:10) light–dark cycles, we obtained continuous measures of energy intake, energy expenditure, and locomotor activity (LMA) over several days using indirect calorimetry. We found that, unlike control mice which exhibited typical nocturnal feeding behavior, DMH[LepR]-silenced mice exhibited a rapid (i.e., within 1 week; *Figure 3—figure supplement 1C–D*) and permanent phase advance in daily food intake (*Figure 3A*), such that dark-cycle food intake was less than light-cycle intake (*Figure 3B*). Similarly, while control mice displayed the expected increase in dark-cycle LMA, this pattern became undetectable within the first week after viral microinjection in DMH[LepR]-silenced mice (*Figure 3C–D*; *Figure 3—figure supplement 1G–H*). Rhythms in other metabolic parameters were similarly shifted and blunted by DMH[LepR] inactivation. Specifically, we found that heat production in DMH[LepR]-silenced mice was reduced selectively in the dark cycle (*Figure 3E–F*) and respiratory-exchange ratio (RER) was elevated in the light cycle (*Figure 3G–H*). These metabolic responses likely reflect the shift of a substantial fraction of daily caloric intake from the dark to the light cycle in DMH[LepR]-silenced mice (*Figure 3A–B*). Our finding that the rapid increase of RER following viral microinjection (*Figure 3—figure supplement 1E–F*) coincides with the timing of excess fat accumulation (*Figure 1F*; *Figure 3—figure supplement 1B*) suggests that DMH[LepR] inactivation may also increase de novo lipogenesis, as is reported, following AgRP neuron activation (*Garfield et al., 2016*; *Cavalcanti-de-Albuquerque et al., 2019*).

Together, these findings suggest that DMH[LepR] neuron activity is required for the normal coupling of daily rhythms in feeding, LMA, and associated metabolic parameters to the light–dark cycle. Whether and how DMH[LepR] neuron activity influences these parameters in the absence of a normal light–dark cycle (i.e., in constant darkness) or in alternate light–dark schedules (e.g., 12:12 vs. 14:10 employed here) are important unanswered questions.

## Female DMH[LepR]-silenced mice recapitulate weight gain and disrupted diurnal rhythmicity seen in males

Although female DMH[LepR]-silenced mice did not exhibit the transient hyperphagia observed in males (*Figure 1—figure supplement 2A–B*), they nonetheless developed a similar degree of obesity (*Figure 1—figure supplement 2C–D*) that was associated with disrupted diurnal rhythms in food intake (*Figure 3—figure supplement 2A–B*), LMA (*Figure 3—figure supplement 2C–D*), heat production (*Figure 3—figure supplement 2E–F*), and RER (*Figure 3—figure supplement 2G–H*), similar to responses observed in male DMH[LepR]-silenced mice. The key role for DMH[LepR] neurons in diurnal behavioral and metabolic control identified in males, therefore, extends to females as well. Given that, compared to male mice (*Chao et al., 2011*), female mice are protected from both hyperphagia and disrupted circadian rhythms during high-fat diet (HFD) feeding (*Palmisano et al., 2017*), future studies are warranted to determine both whether sensitivity to HFD requires DMH[LepR] activity and, if

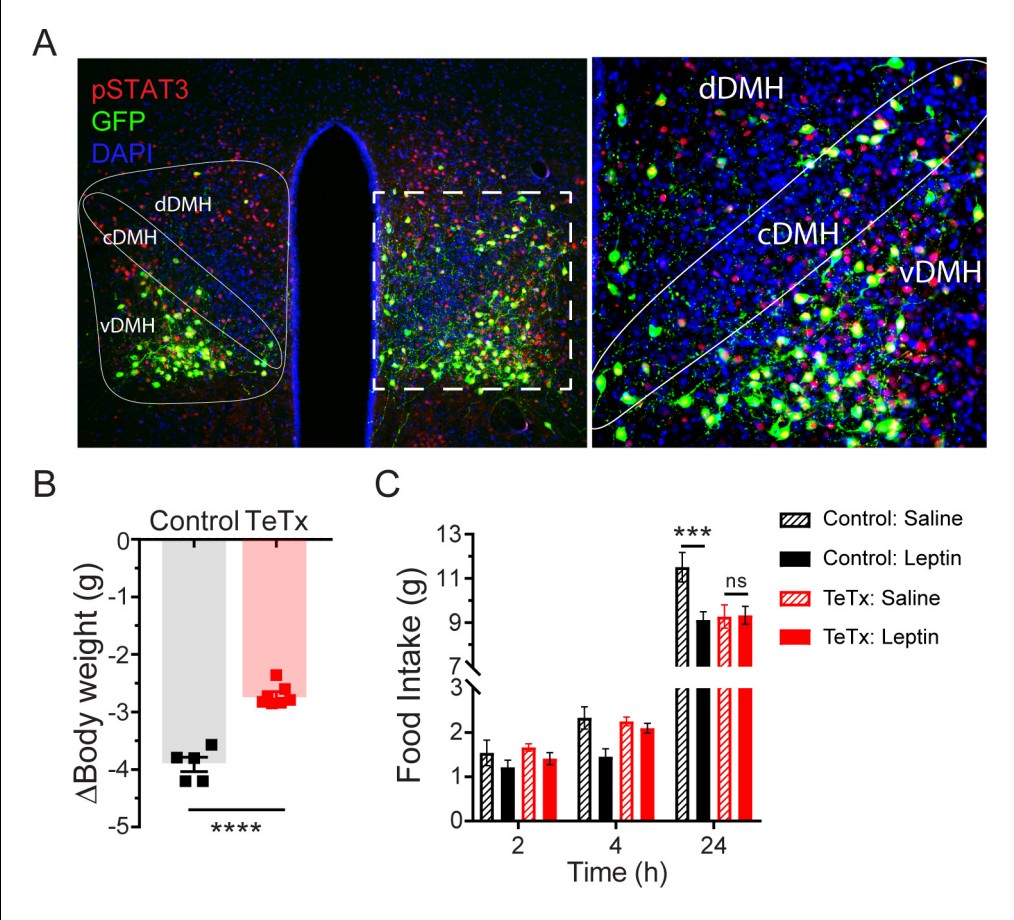

**Figure 2.** Validation of DMH$^{LepR}$ neuronal targeting and evidence that activation of these neurons is required for leptin-induced anorexia. (**A**) Left: Representative image showing extensive overlap of pSTAT3 expression in GFP: TeTx-expressing DMH$^{LepR}$ in mice sacrificed 90 min after leptin administration (i.p. 5 mg/kg). Right: Higher magnification view of the boxed region from the left. (**B**) Change in body weight (unpaired t-test, t=8.483, p=0.0001) following a 24 hr (ZT2–ZT2') fast 5 weeks following viral microinjection and before food was returned in (**C**). (**C**) Post-fast (24 hr) refeeding following i.p. injection of saline or leptin (3 mg/kg). Two-way ANOVA: $F_{(1,4)}$=47.33; p=0.0023 (controls, main effect of leptin). $F_{(1,6)}$=0.1203; p=0.7405 (TeTx, main effect of leptin). v-, c-, and dDMH = ventral, central, and dorsal compartments of the DMH, respectively. Data are mean ± SEM. For repeated measures, post hoc, Sidak's test at each time point is indicated on the graph. *p<0.05, ***p<0.001, ****p<0.0001.

The online version of this article includes the following source data for figure 2:

**Source data 1.** Fast-refeeding +/- Leptin.

so, whether these neurons lie downstream of circuits mediating sexually dimorphic responses to HFD.

## Silencing DMH$^{LepR}$ neurons prevents behavioral adaptation to restricted feeding

To investigate both the extent to which diurnal metabolic disruption in DMH$^{LepR}$-silenced mice is secondary to the shift in daily patterns of food intake and whether DMH$^{LepR}$ neurons are required to entrain feeding behavior, we implemented a time-restricted feeding (TRF) paradigm. To minimize baseline differences in satiety status that could influence TRF adaptation, control and DMH$^{LepR}$-silenced mice were fasted for 24 hr, from ZT14 (dark-cycle onset) until ZT14' of the following day. We then restricted food availability to the dark-cycle, active period (ZT14–ZT24) in both groups. After a 5-day TRF acclimation period, both groups were subjected to four additional days of TRF

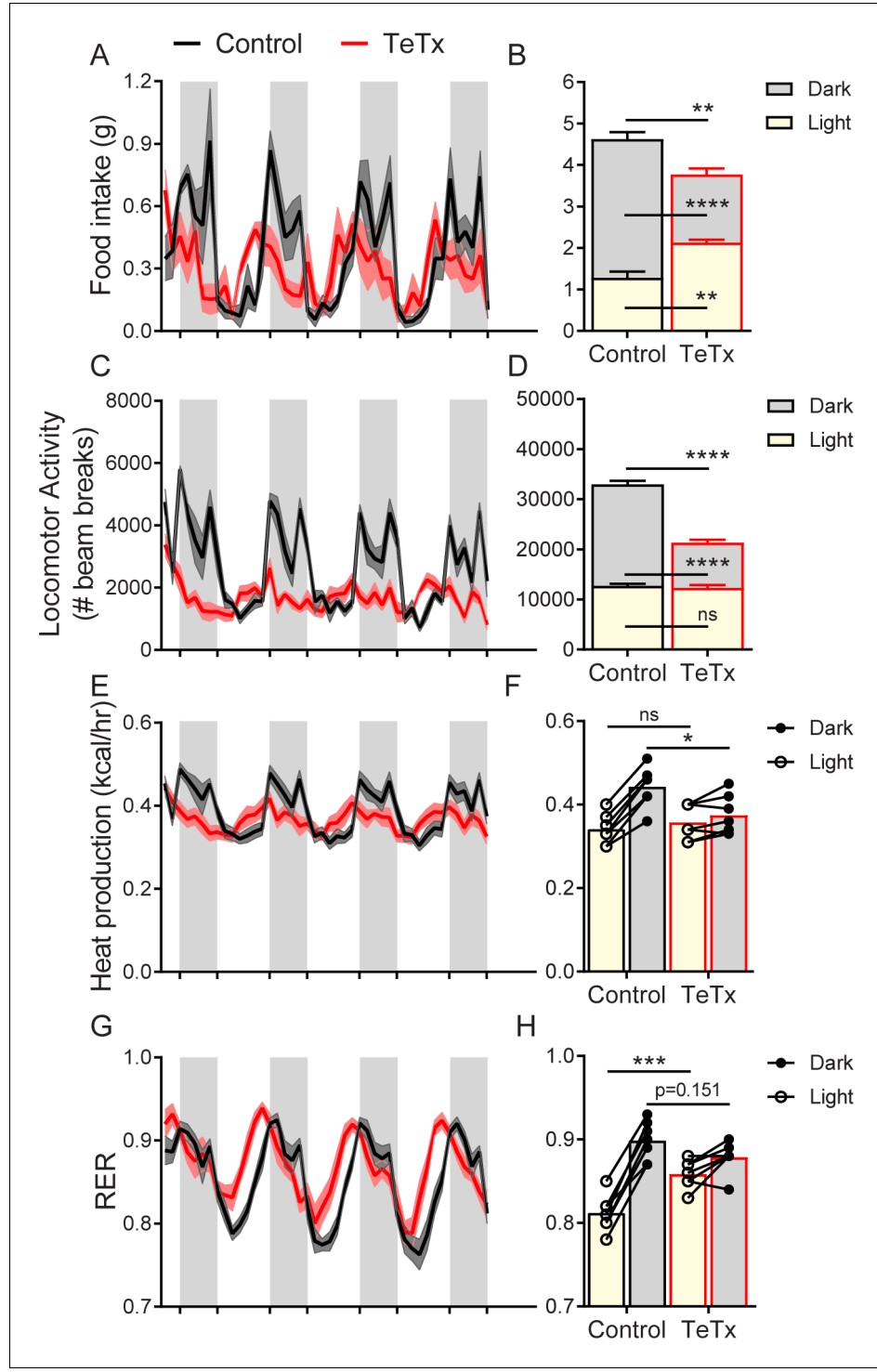

**Figure 3.** DMH$^{LepR}$ neuron inactivation disrupts diurnal patterns of food intake, LMA, heat production, and substrate utilization. Two-hour binned continuous measures (left panels) and mean values across the light (L) and dark (D) periods (right panels) 30 days following microinjection of GFP:TeTx (TeTx; n=7) or GFP control (control; n=7) to the dorsomedial hypothalamic nucleus (DMH) of LepR-Cre+ male mice. Shaded areas indicate dark cycle (ZT14 – ZT24). (**A**) Food intake. Two-way ANOVA: $F_{(1,12)}$=12; p=0.0047 (main effect of TeTx). $F_{(87,1044)}$=2.354; p<0.0001 (time x TeTx interaction). (**B**) Mean food intake from (A) during L, D, and 24-hr periods. Two-way ANOVA: $F_{(1,12)}$=9.567; p=0.0093 (main effect of TeTx). (**C**) Locomotor activity (LMA). Two-way ANOVA: $F_{(1,12)}$=93.22; p<0.0001 (main effect of TeTx). (**D**) Mean LMA from (C) during L, D, and 24-hr periods. Two-way ANOVA: $F_{(1,12)}$=110.4; p<0.0001 (main effect of TeTx). (**E**) Heat production. Two-way ANOVA: $F_{(1,12)}$=1.006;

*Figure 3 continued on next page*

*Figure 3 continued*

p=0.3357 (main effect of TeTx). (**F**) Mean heat production from (**E**) during L and D periods. Two-way ANOVA: $F_{(1,12)}=1.209$; p=0.2930 (main effect of TeTx). (**G**) Respiratory exchange ratio (RER). Two-way ANOVA: $F_{(1,12)}=2.789$; p=0.1208 (main effect of TeTx). (**H**) Mean RER from (**G**) during L and D periods. Two-way ANOVA: $F_{(1,12)}=2.04$; p=0.1788 (main effect of TeTx). Data are mean ± SEM. For repeated measures, post hoc, Sidak's test at each time point is indicated on the graph. *p<0.05,**p<0.01, ***p<0.001, ****p<0.0001.

The online version of this article includes the following source data and figure supplement(s) for figure 3:

**Source data 1.** Female calorimetry.
**Figure supplement 1.** Silencing DMH$^{LepR}$ neurons rapidly and robustly disrupts diurnal rhythms in food intake, peripheral substrate utilization, and LMA.
**Figure supplement 2.** Silencing DMH$^{LepR}$ neurons in female mice recapitulates the effect in males to disrupt diurnal rhythms.

during which measurements were made using indirect calorimetry (for a total of 9 days of TRF), followed by 3 days of ad lib feeding (**Figure 4A**).

During TRF acclimation, body weight oscillated daily as expected in both groups, being higher after food was available during the dark cycle, and lower after light-cycle fasting. However, whereas control mice were able to maintain their weight during TRF by increasing dark-cycle food intake,

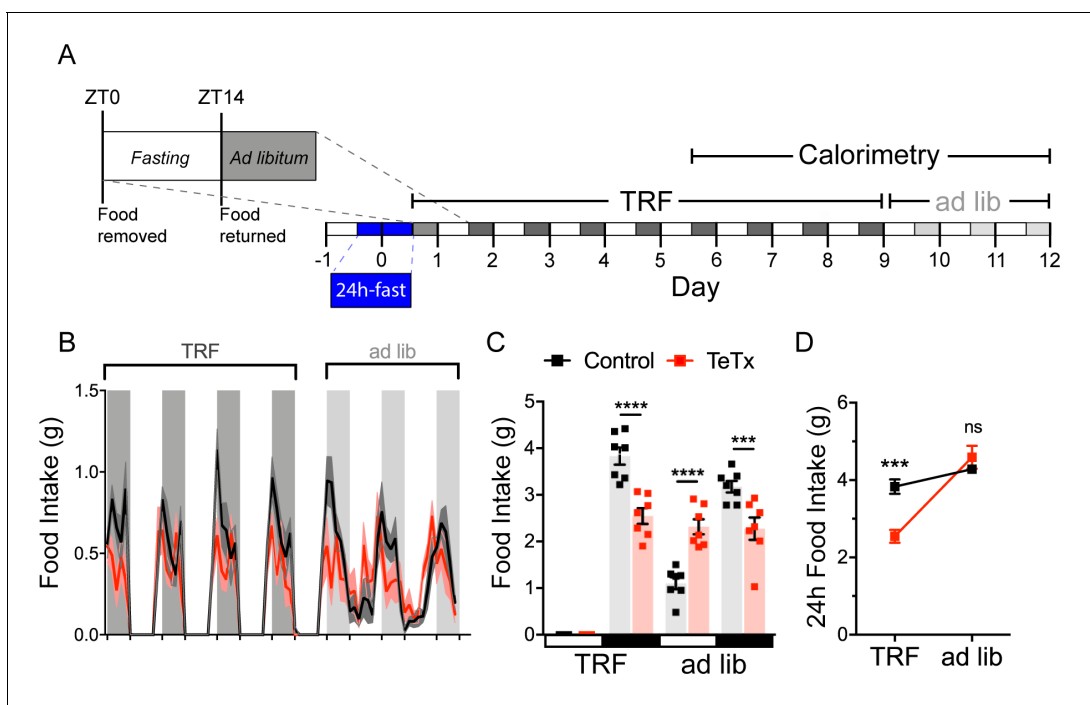

**Figure 4.** DMH$^{LepR}$ neurons are required for adaptation to a dark-cycle restricted feeding schedule. (**A**) Experimental timeline. Six weeks following bilateral microinjection of Cre-dependent GFP:TeTx (TeTx; n=7) or GFP control (control; n=7) to the dorsomedial hypothalamic nucleus (DMH) of LepR-Cre+ male mice, mice were acclimated to time-restricted feeding (TRF) in their home cages for a 5-day lead-in before transfer into direct calorimetry. TRF was maintained in calorimetry for an additional 4 days, followed by ad lib feeding. (**B**) Two-hour binned continuous measures of food intake during TRF and transition back to ad lib feeding. Shaded areas indicate dark cycle (ZT14–ZT24). (**C**) Mean L:D food intake from (**B**) under TRF and ad lib feeding. Two-way ANOVA: $F_{(1,12)}=5.084$; p=0.0436 (main effect of TeTx); $F_{(3,36)}=27.91$; p<0.0001 (time x TeTx interaction). (**D**) Mean 24-hr food intake from (**C**) during TRF and ad lib feeding. Two-way ANOVA: $F_{(1,12)}=5.097$; p=0.0434 (main effect of TeTx); $F_{(1,12)}=47.8$; p<0.0001 (main effect of TRF); $F_{(1,12)}=19.58$; p=0.0008 (TRF x TeTx interaction). Within treatment comparison (TRF vs. ad lib): control $t_{(12)}=1.759$; p=0.1971; TeTx $t_{(12)}=8.018$; p<0.0001. Data are mean ± SEM. For repeated measures, post hoc, Sidak's test at each time point is indicated on the graph. *p<0.05,**p<0.01, ***p<0.001, ****p<0.0001.

The online version of this article includes the following source data and figure supplement(s) for figure 4:

**Source data 1.** Body weight, RER, EE, and LMA during TRF.
**Figure supplement 1.** TRF corrects phase shifts in RER, but has no effect on LMA.

DMH$^{LepR}$-silenced mice failed to compensate for the imposed light-cycle fast and consequently exhibited a small reduction in body weight (*Figure 4—figure supplement 1B*). Upon restoration of ad lib feeding, DMH$^{LepR}$-silenced mice exhibited rebound hyperphagia sufficient to recover lost weight (*Figure 4B–D*), but this hyperphagic response was limited to the light cycle, as DMH$^{LepR}$-silenced mice rapidly reverted to their mistimed feeding rhythms (*Figure 4B–C*). Based on these findings, we conclude that DMH$^{LepR}$ neuron activity is required to entrain feeding behavior during dark-cycle TRF. In addition, the capacity to increase dark-cycle intake in response to weight loss appears to depend upon DMH$^{LepR}$ neuron activity, since DMH$^{LepR}$-silenced mice exhibit rebound hyperphagia following TRF only during the light cycle, a time when normal mice consume little food. Although mechanisms underlying this adaptive response await further study, the capacity to increase intake when food is available for a restricted window each day requires the ability to anticipate when food will be available, in association with a variety of metabolic and neuroendocrine adaptations (*Drazen et al., 2006*). Future studies are warranted to evaluate the extent to which DMH$^{LepR}$ neuronal activity is required for adaptation to an equivalent feeding window during the light cycle, and/or narrower periods of food availability.

## Conclusion

Our work identifies a crucial physiological role for DMH$^{LepR}$ neurons in diurnal patterning of feeding behavior, locomotion, and associated metabolic parameters with normal light–dark cycles, as well as the ability to adapt food intake to a restricted feeding paradigm. Given evidence from both humans and rodents that mistimed feeding can predispose to obesity and T2D (*Challet, 2019*; *Huang et al., 2011*), these findings have relevance to the pathogenesis of both disorders. An improved understanding of the neural circuits underlying endogenous rhythms of behavior, feeding, and metabolism may facilitate the development of new therapeutic and dietary strategies for the treatment of obesity and related metabolic disorders in humans.

# Materials and methods

**Key resources table**

| Reagent type (species) or resource | Designation | Source or reference | Identifiers | Additional information |
|---|---|---|---|---|
| Genetic reagent (*Mus musculus*) | B6; *Lepr*$^{IRES-Cre/+}$ | Jackson Labs | RRID:IMSR_JAX:008320 | |
| Antibody | Anti-GFP (chicken polyclonal) | Abcam | Cat# ab13970; RRID:AB_300798 | IF (1:10,000) |
| Antibody | Anti-pSTAT3 (rabbit monoclonal) | Cell Signaling Technology | Cat# 9145; RRID:AB_2491009 | IF (1:300) |
| Recombinant DNA reagent | AAV1-CBA-DIO-GFP:TeTx | A gift from Richard Palmiter and Larry Zweifel, *Han et al., 2015* | NA | |
| Recombinant DNA reagent | AAV5-hSyn-DIO-EGFP | AddGene | Addgene viral prep #50457-AAV5; RRID:Addgene_50457 | pAAV-hSyn-DIO-EGFP was a gift from Bryan Roth |
| Sequence-based reagent | LepR_WT_forward primer | Jackson Labs | PCR primer | For: 5'- TGCACATTCCCAGCCCAGTGT |
| Sequence-based reagent | Lepr_forward primer | Jackson Labs | PCR primer | For: 5' - CACGACCAAGTGACAGCAAT |
| Sequence-based reagent | Lepr_common_reverse primer | Jackson Labs | PCR primer | Rev: 5' - GACAGGCTCTACTGGAATGGA |
| Peptide, recombinant protein | Recombinant mouse leptin | A F Parlow; National Hormone and Peptide Program | Leptin | |
| Commercial assay or kit | Mouse leptin ELISA | Crystal Chem Cat #90030 | RRID:AB_2722664 | |

*Continued on next page*

*Continued*

| Reagent type (species) or resource | Designation | Source or reference | Identifiers | Additional information |
|---|---|---|---|---|
| Commercial assay or kit | Mouse insulin ELISA | Crystal Chem Cat #90080 | RRID:AB_2783626 | |
| Software, algorithm | Prism 9 | GraphPad | RRID:SCR_002798 | |
| Software, algorithm | ImageJ | Fiji | RRID:SCR_002285 | |
| Software, algorithm | Illustrator | Adobe | RRID:SCR_010279 | |

## Mice

All procedures were performed in accordance with the National Institutes of Health Guide for the Care and Use of Laboratory Animals and were approved by the Animal Care Committee at the University of Washington. Following stereotaxic surgery, all studied animals were individually housed with ad lib access to standard chow diet (LabDiet 5053) in a temperature- and humidity-controlled facility with 14:10 light–dark cycles. Adult *Lepr*$^{IRES-Cre/+}$ (LepR-Cre) mice (Jackson Laboratory no. 008320) or Cre-littermate controls were used for all experiments, as described below.

## Order of experiments

In one cohort of male LepR-Cre (TeTx; n=7) and Cre-littermate (control; n=5) mice, longitudinal measurements of body weight, food intake, and body composition were assessed first (*Figure 1B–H* and *Figure 1—figure supplement 1*), followed by indirect calorimetry (data available upon request), and then fast–refeeding studies with leptin administration (*Figure 2B–C*). A second cohort of male LepR-Cre animals receiving TeTx (n=6) was used to validate the ability of leptin to induce pSTAT3 signaling in DMH$^{LepR}$ neurons (*Figure 2A*). In a third cohort of male LepR-Cre mice (TeTx; n = 7) using GFP as a control (control; n=7), the effect of DMH$^{LepR}$ silencing to induce hyperphagia and weight gain was confirmed (*Figure 3—figure supplement 1*, panels A and B) and the mice were subjected to indirect calorimetry 48 hr after surgery (*Figure 3—figure supplement 1*, panels C–H) and again after 4 weeks (*Figure 3*), which was followed by TRF studies (*Figure 4* and *Figure 4—figure supplement 1*). Finally, a fourth cohort of female LepR-Cre (TeTx; n=8) and Cre-littermate (control; n=8) were used to generate the data shown in *Figure 1—figure supplement 2*, and *Figure 3—figure supplement 2*.

## Stereotaxic surgeries

The viral vector AAV1-CBA-DIO-GFP:TeTx (TeTx) was generated and validated as described (*Han et al., 2015*; *Chen et al., 2018*) and generously provided by Dr. Richard Palmiter and Dr. Larry Zweifel (University of Washington, Seattle, WA). As a control in some studies (see 'Order of Experiments'), the viral vector AAV5-hSyn-DIO-EGFP (AddGene, Watertown, MA; cat: 50457-AAV5) was used. For viral microinjection, animals were placed in a stereotaxic frame (Kopf 1900; Cartesian Research Inc, Tujunga, CA) under isoflurane anesthesia. The skull was exposed with a small incision, and two small holes were drilled for bilateral 200 nL injection volume of TeTx into the DMH of LepR-Cre or Cre-negative littermate mice based on coordinates from the Mouse Brain Atlas: anterior-posterior (AP) −1.6, dorsal-ventral (DV) −5.6 mm, and lateral 0.40 mm from bregma, identified as the approximate point at which coronal and saggital sutures intersect, and where the bregma–lambda distance approximates 4.21 mm, as previously described (*Faber et al., 2020*; *Franklin and Paxinos, 2008*). AAV) was delivered using a Hamilton syringe with a 33-gauge needle at a rate of 50 nL/min (Micro4 controller), followed by a 5 min wait at the injection site and a 1 min wait 0.05 mm dorsal to the injection site before needle withdrawal. Animals received a perioperative subcutaneous injection of buprenorphine hydrochloride (0.05 mg/kg; Reckitt Benckiser, Richmond, VA). Viral expression was verified post hoc in all animals, and any data from animals in which the virus expressed outside the targeted area were excluded from the analysis.

## Body composition analysis

Measurements of body lean and fat mass were determined in live, conscious mice by use of quantitative magnetic resonance spectroscopy (QMR; EchoMRI 3-in-1; Echo MRI, Houston, TX) by the University of Washington Nutrition Obesity Research Center Energy Balance Core.

## Leptin effects on food intake and pSTAT3 induction

To validate the ability of leptin to elicit pSTAT3 signaling in DMH$^{LepR}$ neurons, ad lib fed mice were injected intraperitoneally with leptin (5 mg/kg; Dr. Parlow; National Hormone Peptide Program) and perfused 90 min later, as described below.

To assess the ability of leptin to suppress the compensatory hyperphagia that normally follows a prolonged fast, mice were fasted for 24 hr from ZT2 to ZT2'. On the second day, leptin (3 mg/kg) or vehicle control (PBS, pH 7.9) was injected intraperitoneally in mice 15 min before preweighed food was placed back in the cage, and intake was monitored for the following 24 hr.

## Indirect calorimetry, food intake, and activity

Prior to surgery, mice were acclimated to calorimetry cages for at least 7 days, during which time baseline pre-intervention calorimetry values were obtained for each subject. To account for potential confounding effects of stress resulting from rehousing into calorimetry during study, the first 12–16 hr of data (including the first dark cycle) were omitted from all analyses, except for the TRF experiment where continuous intake measurements were required. Energy expenditure measurements were obtained by a computer-controlled indirect calorimeter system (Promethion, Sable Systems, Las Vegas NV) with support from the Energy Balance Core of the NORC at the University of Washington, as previously described (*Kaiyala et al., 2015*). Oxygen consumption (VO$_2$) and carbon dioxide production (VCO$_2$) were sequentially measured for each mouse for 30 s at 5 min intervals, and food and water intakes were measured continuously while mice were housed in a temperature- and humidity-controlled cabinet (Caron Products and Services, Marietta, OH). Ambulatory activity was determined simultaneously and beam breaks in the y-axis were scored as an activity count, and a tally was recorded every 3 min. Data acquisition and instrument control were coordinated by MetaScreen v2.3.15.11, and raw data were processed and binned into 2-hr increments using ExpeData v1.9.14 (Sable Systems, Las Vegas, NV) using an analysis script documenting all aspects of data transformation.

## Time-restricted feeding

To eliminate the initial effects of varying fed status of animals, 1 day before TRF, animals were placed into clean bedding and fasted for 24 hr from ZT14 (on day 1) to ZT14' (on day 0) before TRF began. Food was removed each morning at the start of the light cycle (ZT0) and the bedding inspected to ensure no residual food debris remained accessible. Food was returned at the start of the dark cycle (ZT14) each day. Body weight was also measured at both ZT0 and ZT14 daily. Animals were maintained on TRF for a total of 5 days in home cages, then subjected to indirect calorimetry for four additional days of TRF before returning to ad lib feeding for the remaining 3 days of study (*Figure 4A*).

## Immunohistochemistry

For brain immunohistochemical analyses, animals were terminally anesthetized with ketamine:xylazine and transcardially perfused with PBS followed by 4% paraformaldehyde in 0.1 mol/L PBS. Brains were removed and postfixed overnight, then transferred into 30% sucrose overnight or until brains sunk in solution. Brains were subsequently sectioned on a freezing-stage microtome (Leica) to obtain 30 µm coronal sections in four series. A single series of sections per animal was used in histological studies, and the remainder stored in −20°C in cryoprotectant. Brain sections were washed in PBS with Tween-20, pH 7.4 (PBST) overnight at 4°C. Sections were then washed at room temperature in PBST (3 x 8 min), followed by a blocking buffer (5% normal donkey serum [NDS], 1% bovine serum albumin [BSA] in PBST with azide) for 60 min with rocking. Sections were then incubated overnight at 4°C in blocking buffer containing primary antiserum (goat anti-GFP, Fitzgerald, 1:1000; rabbit anti-pSTAT3, Sigma-Aldrich, St Louis, MO, 1:1000). Next, sections were washed (3 x 8 min) in PBST before incubating in secondary donkey anti-goat IgG Alexa 488 (Jackson ImmunoResearch

Laboratories, West Grove, PA) diluted 1:1000 in blocking buffer. Sections were washed (3 x 8 min) in PBST before incubating with DAPI for 8 min, followed by a final wash (3 x 10 min) in PBS. Sections were mounted to slides and imaged using a Leica SP8X confocal.

## Tissue processing and blood collection

Tail blood for plasma hormonal measurement was collected at indicated times. Blood was collected via EDTA-coated capillary tubes and centrifuged at 4°C (7000 rpm, 4 min), and plasma was subsequently removed and stored at −80°C for subsequent assay. Plasma leptin (Crystal Chem, Elk Grove Village, IL; #90030) and plasma insulin (Crystal Chem; #90080) were determined by ELISA.

## Statistical analyses

Prior to analysis, all data were tested for normality by Shapiro–Wilk normality test. All results are presented as means ± SEM. $P$ values for unpaired comparisons were calculated by two-tailed Student's $t$ test. Time course comparisons between groups were analyzed using a two-way repeated-measures ANOVA with main effects of treatment (control vs. TeTx) and time. All post hoc comparisons were determined using Sidak's correction for multiple comparisons. All statistical tests indicated were performed using Prism (version 7.4; GraphPad, San Diego, CA) software.

## Acknowledgements

We thank R Palmiter, C Campos, L Zweifel, and M Baird for producing the TeTx virus, JT Nelson for assistance with metabolic experiments, and V Damian for maintaining the mouse colony. We also thank R Palmiter for editing the manuscript. We are grateful to N Peters at the University of Washington Keck Imaging Center for technical assistance and the National Institutes of Health (S10-OD-016240) for support to the W.M. Keck Foundation Center for Advanced Studies in Neural Signaling. This work was supported by NIH grants F31-DK113673 (CLF), T32-GM095421 (CLF), DK128802 (ZM), DK089056 and DK124238 (GJM), DK083042 and DK101997 (MWS); the NIDDK-funded Nutrition Obesity Research Center (DK035816) and Diabetes Research Center (DK017047) and the Diabetes, Obesity, and Metabolism (T32 DK007247; CLF) and Nutrition, Obesity, and Atherosclerosis (T32 HL007028; JDD) training grants at the University of Washington; a Department of Defense CDMRP/PRMRP grant W81XWH2010250 (ZM); a Dick and Julia McAbee Endowed Fellowship (JDD); an American Diabetes Association Innovative Basic Science Award (ADA 1–19-IBS-192; GJM); and an American Diabetes Association fellowship grant (ADA 1–19-PDF-103; JDD).

## Additional information

### Funding

| Funder | Grant reference number | Author |
|---|---|---|
| National Institutes of Health | F31-DK113673 | Chelsea L Faber |
| National Institutes of Health | T32-GM095421 | Chelsea L Faber |
| National Institutes of Health | DK089056 | Gregory J Morton |
| National Institutes of Health | DK124238 | Gregory J Morton |
| National Institutes of Health | DK083042 | Michael W Schwartz |
| National Institutes of Health | DK101997 | Michael W Schwartz |
| National Institutes of Health | T32 DK007247 | Chelsea L Faber |
| National Institutes of Health | T32 HL007028 | Jennifer D Deem |
| American Diabetes Association | ADA 1-19-IBS-192 | Gregory J Morton |
| American Diabetes Association | ADA 1-19-PDF-103 | Jennifer D Deem |
| U.S. Department of Defense | W81XWH2010250 | Zaman Mirzadeh |
| National Institutes of Health | DK128802 | Zaman Mirzadeh |
| NIDDK | DK035816 | Michael W Schwartz |

| University of Washington | Dick and Julia McAbeeEndowed Fellowship | Jennifer D Deem |
| NIDDK | DK017047 | Jennifer D Deem |

The funders had no role in study design, data collection and interpretation, or the decision to submit the work for publication.

### Author contributions

Chelsea L Faber, Conceptualization, Data curation, Formal analysis, Validation, Investigation, Visualization, Methodology, Writing - original draft, Writing - review and editing; Jennifer D Deem, Investigation, Writing - review and editing; Bao Anh Phan, Investigation; Tammy P Doan, Validation, Investigation; Kayoko Ogimoto, Software, Investigation; Zaman Mirzadeh, Michael W Schwartz, Conceptualization, Supervision, Writing - review and editing; Gregory J Morton, Conceptualization, Resources, Supervision, Funding acquisition, Methodology, Project administration, Writing - review and editing

### Author ORCIDs

Chelsea L Faber https://orcid.org/0000-0002-4812-8164
Jennifer D Deem http://orcid.org/0000-0002-8865-5145
Michael W Schwartz http://orcid.org/0000-0003-1619-0331
Gregory J Morton https://orcid.org/0000-0002-8106-8386

### Ethics

Animal experimentation: All procedures were performed in accordance with the National Institutes of Health Guide for the Care and Use of Laboratory Animals and were approved by the Animal Care Committee at the University of Washington. (Jackson Laboratory no. 008320).

### Decision letter and Author response

Decision letter https://doi.org/10.7554/eLife.63671.sa1
Author response https://doi.org/10.7554/eLife.63671.sa2

## Additional files

### Supplementary files

• Transparent reporting form

### Data availability

All data generated or analysed during this study are included in the manuscript and supporting files. Source data files have been provided for Figures 1–4.

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
