## [Decision Letter]

**Acceptance summary:**

The interactions of brain pathways regulating circadian patterns and metabolism is an important area of study and will be of wide interest to the readers of *eLife*. Your work also provides a novel role for leptin signaling in the dorsal medial nucleus of the hypothalamus.

**Decision letter after peer review:**

Thank you for submitting your article "LepR neurons in the dorsomedial hypothalamus regulate the timing of circadian rhythms in feeding and metabolism" for consideration by *eLife*. Your article has been reviewed by three peer reviewers, one of whom is a member of our Board of Reviewing Editors, and the evaluation has been overseen by Kate Wassum as the Senior Editor. The reviewers have opted to remain anonymous.

The reviewers have discussed the reviews with one another and the Reviewing Editor has drafted this decision to help you prepare a revised submission.

Summary:

Faber and colleagues describe results from a series of studies investigating the effects of knocking down leptin receptor expression in the dorsal medial nucleus (DMH) of the hypothalamus. They report that silencing of leptin receptor expression induces increased weight gain and adiposity, a shift in timing of daily food intake and associated metabolic rhythms, and decreased activity and heat production. Specifically, they show that chronic inactivation of DMH-LepR+ neurons results in mistimed feeding under light:dark conditions and bouts of hyperphagia leading to increased body weight. Conversely, loss of the DMH-LepR+ results in less acute weight loss upon 24-hr fasting and elimination of leptin responsive rebound feeding with this fast. Lastly, the animals are unable to entrain to food provision under a time-restricted feeding schedule with access limited to the "dark" period. Their work suggests a role for LepR-expressing DMH cell types in the relationship between feeding rhythm and homeostatic energy signaling. The findings implicate these neurons in mediating food anticipatory activity. Overall, the studies are interesting and are potentially of wide interest. However, several technical issues need to be addressed.

Essential revisions:

1) It has been reported that AAV1 has the capability for anterograde transport (https://www.sciencedirect.com/science/article/pii/S0896627316309138). This may limit/temper the interpretations as AAV1-CBA-DIO-GFP:TeTX will likely silence other LepR-Cre expressing neuronal groups which are downstream of DMH^LepR^ neurons. If these downstream are LeprCre-expressing POMC or other critical hypothalamic neurons, the body weight phenotype or leptin-induced anorexic effects may come from the silencing of downstream LeprCre-expressing neurons, not DMH^LepR^ neurons themselves. Whether this occurred and its impact on the results needs clarification.

2) Given that different serotypes and are more or less able to spread and (also based on different promoters) infect anatomical regions and their neurons, maintaining serotype and the promoter type would have reduced the risk that the results are due to differences in viral characteristics and thus neuronal targeting efficiency. Typically, the AAV5 serotype that was used for controls produces much more sparse targeting in the hypothalamus than serotypes such as 1, 8, 9 or DJ. The authors only show data from the AAV1 group (AAV1-CBA-DIO-GFP:TeTx) but not from the AAV5 GFP control vector. It would be good to show this. Furthermore, it may be good to show weight trajectories for Cre positive and Cre negative animals, given that these animals are used for the data presented in Figure 1 (there, both groups are injected with the same vector, but the effect of Cre recombinase cannot be excluded). Finally, while stereotaxic coordinates are provided for the viral injections, Bregma is not defined (the definition of this can differ and defining this will facilitate replication of the current data).

3) The virus tracing and discussion raises some question as to specificity as cell body GFP patterns appear to be present within the arcuate following injections-is it possible that some virus uptake has occurred within LepR-Cre positive cells outside of DMH (as seen in the figure inset and Figure 1—figure supplement 1)? It would be good to provide a quantification of GFP-positive cell bodies within the DMH vs. ARC for the included animals. Furthermore, with the leptin treatment, was activity of pSTAT3 observed (or different) in other sites?

4) The authors don't directly demonstrate whether TeTx approaches silence of DMH^LepR^ neurons, e.g., by electrophysiology or c-Fos induction after leptin administration. This should be addressed.

5) The most profound circadian disruption in feeding occurs when mice are subject to high-fat diet. In humans, energy-dense diet is also "believed" to drive night time eating. Some people also suffer from night-eating syndrome in which they tend to eat mostly at night. One of the forms of night eating syndrome is observed in Smith-Magenis syndrome. The manuscript uses mice fed a normal chow. But it leaves open the question of whether the DMH-LepR neurons are involved in the HFD induced circadian feeding rhythm disruption or the results are just epiphenomenon of acute disorder of a neuronal circuit implicated in feeding. Without addressing this question, the manuscript leaves open the question of whether the DMH-LepR neurons are, in fact, the actual mediators of circadian feeding rhythm as they relate to disrupted physiology found in ad lib fed mice on HFD or in free-living human. At a minimum, this needs to be discussed.

6) The TRF and ad lib experiment in Figure 4 is difficult to interpret. The experiment protocol of adapting mice to just 5 days of TRF and releasing them to ad lib feeding may not be sufficient for mice to adapt to a TRF condition. Typically, high-fat diet fed mice who consume ~30% food at night can take from 1- 2 weeks to adapt to TRF and consume an isocaloric diet as ad lib feeding "prior to TRF". In other words, do the DMH-LepR neuron function of ad lib fed mice on HFD resemble DMH-LepR silenced mice?

7) On a separate note, mice may also need more than 2 days to adapt to indirect calorimetry before their RER, and other metabolic parameters reach equilibrium. Since this is a crucial experiment that related to the conclusion and title of the manuscript, the experiment protocol and interpretation should be conducted to highest possible standard leaving no chance for alternate interpretation.

8) Circadian rhythm analysis formally requires examination under constant conditions (darkness) in order to observe "free running" endogenous rhythms vs. entrainment to light. Here, there appears to be 14/10 light cycle imposed throughout-so the analyses are that of "diurnal" behaviors-this is a key consideration in the interpretation and discussion throughout (i.e., to replace the term "circadian" with "diurnal"). Was LD 14:10 used throughout?-if so, this may change results especially with respect to entrainment which is usually studied under 12:12 conditions. This needs clarification.

9) The data for 24-hr fasting (Figure 2) indicate less weight loss with fasting upon DMH-LepR+ silencing-in addition to abrogation of rebound feeding. This would seem to indicate that the silenced-cells exert a net activating input into orexigenic (AgRP) cells as their loss reduces feeding drive yet (paradoxically) results in conservation of energy with the 24-hr. fast (i.e., less weight loss). Is DMH-inhibiting an inhibitor such as PVH or POMC vs. directly modulating AgRP (if the latter were the case, one would conclude that DMH is activating)? This needs to be clarified and discussed.

10) With regard to food entrainment and anticipation under the experiments in Figure 4, the results provide intriguing evidence that DMH-LepR neurons play a role in adaptation to food availability within limited time windows under light:dark (diurnal) conditions. Whether this reflects changes in a circadian behavioral process and/or a circadian mechanism is uncertain-the adaptive response is sustained as the authors note even in SCN-ablated animals so the link to clock mechanisms has remained mysterious (this aspect can be included through revision of the text and conclusions rather than new experimentation).

11) When the mice with LepR deletion in the DMH undergo fasting-refeeding, they first exhibit significantly less weight loss (Figure 2B), and then compensate with less rebound feeding, which is not further blunted by leptin (as opposed to the anorexic response to leptin in fasted controls). This initial observation makes it hard to interpret whether the lack of a leptin-mediated effect in TeTx animals is simply due to the attenuated weight loss during fasting in these animals.

---

## [Author Response]

Essential revisions:1) It has been reported that AAV1 has the capability for anterograde transport (https://www.sciencedirect.com/science/article/pii/S0896627316309138). This may limit/temper the interpretations as AAV1-CBA-DIO-GFP:TeTX will likely silence other LepR-Cre expressing neuronal groups which are downstream of DMH_LepR_ neurons. If these downstream are LeprCre-expressing POMC or other critical hypothalamic neurons, the body weight phenotype or leptin-induced anorexic effects may come from the silencing of downstream LeprCre-expressing neurons, not DMH^LepR^ neurons themselves. Whether this occurred and its impact on the results needs clarification.

Although transsynaptic spread of AAV1 has been reported, to our knowledge, this outcome has been largely limited to applications in the cortex; in our hands, we have attempted and failed to achieve transsynaptic spread in hypothalamic brain regions even with very high titers (as determined by AAV1-Cre injection into hypothalamus of Ai14 reporter mice; not shown). To this we add recent evidence that the mechanism for AAV1 transsynaptic spread requires synaptic vesicle release from the starter population (Zingg et al., J Neuro 2020), which TeTx itself precludes, as they show nicely in the paper. Thus, while it remains possible that in our studies a small number of viral particles were trafficked and packaged into synaptic vesicles before TeTx-mediated VAMP2 cleavage, the absence of detectable GFP:TeTx+ cell bodies in the ARC or other known downstream projections of DMH^LepR^ neurons speaks against this possibility. To add clarity to this issue, we now include higher resolution images of the DMH and ARC in which the viral reporter is colocalized with DAPI, in the revised Figure 1C. The images clearly show no overlap between GFP:TeTx+ terminals and ARC nuclei, unlike the pronounced overlap observed in the DMH. Thus, we do not believe that our data are confounded by viral spread to LepR+ neurons located outside of the DMH.

2) Given that different serotypes and are more or less able to spread and (also based on different promoters) infect anatomical regions and their neurons, maintaining serotype and the promoter type would have reduced the risk that the results are due to differences in viral characteristics and thus neuronal targeting efficiency. Typically, the AAV5 serotype that was used for controls produces much more sparse targeting in the hypothalamus than serotypes such as 1, 8, 9 or DJ. The authors only show data from the AAV1 group (AAV1-CBA-DIO-GFP:TeTx) but not from the AAV5 GFP control vector. It would be good to show this. Furthermore, it may be good to show weight trajectories for Cre positive and Cre negative animals, given that these animals are used for the data presented in Figure 1 (there, both groups are injected with the same vector, but the effect of Cre recombinase cannot be excluded).

We thank the reviewer for the suggestions and are happy to provide additional data, as requested. To clarify the experimental design, the original Figure 1 included Cre+ and Cre- animals, each of which were injected with the same AAV1-CMV-DIO-GFP:TeTx vector, and body weight trajectories presented in original Figure 1F. In the revised manuscript, we have included additional data from the second cohort of animals, wherein only Cre+ animals were studied and AAV5-DIO-GFP was delivered as a control – in the revised manuscript, these can be found in a new supplement to Figure 3, entitled: Figure 3—figure supplement 1. We further clarify the animal numbers and controls within the relevant figure legends of the revised manuscript, as well as within the revised Materials and methods (see subsection “Order of Experiments”).

In this figure, in addition to showing the requested targeting validation and body weight trajectories, we also show data from when these animals were placed into calorimetry 48-hr after microinjection surgery to ascertain the timing of feeding and metabolic disruptions relative to changes in body weight and/or adiposity. Although the magnitude of the early hyperphagia is somewhat diminished in this cohort relative to the previous cohort, and failed to achieve statistical significance (perhaps owing to the smaller n and/or the effects of housing mice in calorimetry-relative to home cages), we nonetheless observed the rapid emergence of the phase-advance in food intake following inactivation of DMH^LepR^ neurons using this approach, along with a rapid and robust elevation in respiratory exchange ratio (RER), and diminished dark-cycle locomotor activity (LMA). An expanded discussion of these effects and their interpretation can be found in the Results and Discussion of the revised manuscript.

Finally, while stereotaxic coordinates are provided for the viral injections, Bregma is not defined (the definition of this can differ and defining this will facilitate replication of the current data).

We fully acknowledge the difficulty in surgical targeting where definitions of coordinates may differ, and thank the reviewer for their attention to this important detail. In response, we have clarified our bregma assignment within the Materials and methods (see subsection “Stereotaxic Surgeries”).

3) The virus tracing and discussion raises some question as to specificity as cell body GFP patterns appear to be present within the arcuate following injections – is it possible that some virus uptake has occurred within LepR-Cre positive cells outside of DMH (as seen in the figure inset and Figure 1—figure supplement 1)? It would be good to provide a quantification of GFP-positive cell bodies within the DMH vs. ARC for the included animals. Furthermore, with the leptin treatment, was activity of pSTAT3 observed (or different) in other sites?

As discussed in more detail in response to point 1 above, we do not detect any GFP:TeTx+ cell bodies within the ARC, and we now include higher resolution images with DAPI to add clarity to this issue (see revised Figure 1C). Regarding the level of pSTAT3 activity, we emphasize that TeTx expression does not interfere with intracellular signaling cascades – it simply prevents release of synaptic vesicles. Consequently, we did not expect to see any impairment in leptin-induced activation of pSTAT3 in GFP:TeTx relative to GFP control mice. To expand on this point, we take this opportunity to clarify that we did not delete leptin receptors from DMH neurons in our studies, as is asserted in the Editor’s summary. Had we done so, we would have expected to block leptin-induced pSTAT3 induction, as we have reported previously in studies using a Cre-lox approach to delete leptin receptors from VMN^SF1^ neurons. In that study, we showed that VMN-specific deletion of LepRb eliminated leptin-induced activation of pSTAT3 in the VMN, while having no effect in the ARC (Meek, Endo 2013), and others have reported similar findings (Dhillon, Neuron 2006). The key point of the study, which we emphasize here, was to use leptin-induced pSTAT3 to validate that GFP:TeTx expression was limited to leptin-responsive neurons within the DMH, rather than to show whether TeTx inactivation interferes with leptin-induced pSTAT3 induction, which was not expected and not observed. We have now clarified this point in the Results and Discussion of the revised manuscript.

4) The authors don't directly demonstrate whether TeTx approaches silence of DMH^LepR^ neurons, e.g., by electrophysiology or c-Fos induction after leptin administration. This should be addressed.

While we acknowledge the importance of this concern, we emphasize that this work has already been done by our collaborators Dr. Palmiter and Dr. Zweifel, who generated, characterized, and provided the TeTx viral construct employed in these studies. Please see Han et al., 2015 describing the validation, as well as subsequent use of this virus by others (Campos et al., 2017). These references are now included in the revised manuscript in support of this important issue. To this we add that because TeTx expression blocks neither action potential firing nor Fos induction, validation by electrophysiology requires assessment of the ability of TeTx-expressing neurons to influence post-synaptic currents, and acquiring such data would delay publication by months, as our lab is not equipped to do this type of analysis. We hope the reviewers will agree that prior validation of the construct performed by our collaborators, together with a phenotype that is highly reproducible across multiple cohorts and our use of both Cre+ and Cre- controls, collectively provide strong evidence that the expected blockade of neurotransmission was achieved following DMH^LepR^ TeTx expression.

5) The most profound circadian disruption in feeding occurs when mice are subject to high-fat diet. In humans, energy-dense diet is also "believed" to drive night time eating. Some people also suffer from night-eating syndrome in which they tend to eat mostly at night. One of the forms of night eating syndrome is observed in Smith-Magenis syndrome. The manuscript uses mice fed a normal chow. But it leaves open the question of whether the DMH-LepR neurons are involved in the HFD induced circadian feeding rhythm disruption or the results are just epiphenomenon of acute disorder of a neuronal circuit implicated in feeding. Without addressing this question, the manuscript leaves open the question of whether the DMH-LepR neurons are, in fact, the actual mediators of circadian feeding rhythm as they relate to disrupted physiology found in ad lib fed mice on HFD or in free-living human. At a minimum, this needs to be discussed.

We agree that the possibility that DMH^LepR^ neurons mediate disrupted feeding rhythms associated with the switch to HFD is quite interesting. Indeed, we made this point in the original manuscript, and as this question is a key element of a grant application in preparation, we are glad to know that the reviewer shares our interest in this topic. While we are happy to expand discussion of this issue beyond its identification as a priority for future studies, we are hard-pressed to do so if we are to adhere to mandatory text limits. Should the reviewers and editors feel this to be of sufficient import to waive the text limit requirements, however, we are happy to expand on this point.

6) The TRF and ad lib experiment in Figure 4 is difficult to interpret. The experiment protocol of adapting mice to just 5 days of TRF and releasing them to ad lib feeding may not be sufficient for mice to adapt to a TRF condition.

To clarify the experimental design, as depicted in Figure 4A of the original manuscript, animals were maintained on TRF for a total of 9 days before release to ad lib feeding. We acknowledge this timeline may be confusing to the reader, given that the first 5 days of TRF occurred in home cages, while the latter 4 days occurred in calorimetry housing, and we have endeavored to clarify this point in the relevant description of the methods; further, the missing axis label for Figure 4A has been added (“Days”) in the revised manuscript.

We also agree with the reviewer’s point that the length of TRF acclimation is an important consideration for the interpretation of the findings. Despite the relatively short duration TRF we employed in comparison to some other studies, we emphasize that whereas control mice were able to adapt effectively so as to increase food intake during the dark cycle and thereby avert weight loss, this adaptation did not occur in DMH^LepR^ silenced mice, resulting in failure to maintain body weight. Thus, the TRF protocol was of sufficient duration to reveal a phenotype that we believe is of interest. While it is possible that experimental animals take longer to adapt to dark-cycle TRF, maintaining TRF for longer periods may have introduced its own set of confounding variables (greater weight loss leading to a confounding increase in homeostatic drive to regain lost body weight, for example). It is out of this concern that we opted to examine the body weight and feeding responses using a shorter, rather than prolonged, TRF paradigm to assess behavioral adaptation to this challenge. However, we agree that future studies are warranted to examine adaptation to different TRF paradigms, including equivalent periods throughout the dark-light cycle, a point we highlight in the Results and Discussion.

Typically, high-fat diet fed mice who consume ~30% food at night can take from 1- 2 weeks to adapt to TRF and consume an isocaloric diet as ad lib feeding "prior to TRF". In other words, do the DMH-LepR neuron function of ad lib fed mice on HFD resemble DMH-LepR silenced mice?

We are similarly interested in the potential role played by DMH^LepR^ neurons in the circadian disruption and “mistimed” feeding that occurs following HFD introduction. We hope that the publication of studies included herein will provide a useful foundation for future work (and associated funding) that investigate whether HFD feeding disrupts feeding cycle through these same neuronal subsets. To this end, we hope that the “Short Report” manuscript format will allow timely dissemination of this informative work to the research community and provide a foundation upon which future studies may be built.

7) On a separate note, mice may also need more than 2 days to adapt to indirect calorimetry before their RER, and other metabolic parameters reach equilibrium. Since this is a crucial experiment that related to the conclusion and title of the manuscript, the experiment protocol and interpretation should be conducted to highest possible standard leaving no chance for alternate interpretation.

We acknowledge and are fully aware of the importance of allowing mice to adapt to indirect calorimetry cages prior to study. The studies were performed in the NIDDK-funded Nutrition Obesity Research Center (NORC) Energy Balance Core (Director: Dr. Morton), which has nearly 20 years of experience and has conducted hundreds of these types of studies. In so doing, this facility has developed and implemented steps designed specifically to reduce the metabolic and behavioral impact of re-housing, and to both minimize confounding influences on indirect calorimetry data and recognize them when they occur. To add clarity to the steps taken for studies in the current manuscript, we have added detail to the Materials and methods section (see subsection “Indirect Calorimetry, Food Intake, and Activity”). Specifically, before any surgical intervention (e.g., viral microinjection), all study animals were subjected to calorimetry for at least 7 days both to collect baseline data, and to acclimate animals to experimental conditions. In addition, we routinely omit the first 16-hr block of data (including the entire first dark-cycle) from all analyses (except for the TRF study where continuous measures were necessary), based on prior validation of this step as a way to eliminate confounding effects arising from a change in housing. In support of the efficacy of these quality control measures, we note that relevant data from control mice (food intake, locomotor activity, heat production and RER) are quite reproducible over many successive light and dark cycles; had the animals not yet equilibrated when the study began, the data would have evolved over time, but this was not observed.

8) Circadian rhythm analysis formally requires examination under constant conditions (darkness) in order to observe "free running" endogenous rhythms vs. entrainment to light. Here, there appears to be 14/10 light cycle imposed throughout-so the analyses are that of "diurnal" behaviors-this is a key consideration in the interpretation and discussion throughout (i.e., to replace the term "circadian" with "diurnal"). Was LD 14:10 used throughout?-if so, this may change results especially with respect to entrainment which is usually studied under 12:12 conditions. This needs clarification.

We acknowledge the limitation inherent in studying animals in 14:10 light:dark cycles, and intend in future studies to explore the effects of DMH^LepR^ inactivation on free-running circadian rhythms in mice housed in constant darkness. (This is another project included in a pending grant proposal.) With this objective in mind, we have revised the text to better reflect the primary outcome of disrupted *diurnal* rhythms in association with normal light:dark cycles, and we reserve the term *circadian* to refer to outcomes in environmentally constant conditions. We further clarify the 14:10 light:dark schedule utilized in the Results and Discussion, and suggest that future studies may be needed to clarify whether alternate lighting schedules may influence the phenotype.

9) The data for 24-hr fasting (Figure 2) indicate less weight loss with fasting upon DMH-LepR+ silencing-in addition to abrogation of rebound feeding. This would seem to indicate that the silenced-cells exert a net activating input into orexigenic (AgRP) cells as their loss reduces feeding drive yet (paradoxically) results in conservation of energy with the 24-hr. fast (i.e., less weight loss). Is DMH-inhibiting an inhibitor such as PVH or POMC vs. directly modulating AgRP (if the latter were the case, one would conclude that DMH is activating)? This needs to be clarified and discussed.

We thank the reviewer for raising this issue, which we agree is of interest. Based upon similar comments in point 11 below, we have expanded the Discussion to address the potential contribution made by post-synaptic AgRP neurons in the phenotype resulting from DMH^LepR^ inactivation.

Previous retrograde tracing and electrophysiology data (see Garfield et al., 2016), indicates that a subset of DMH^LepR^ neurons are GABAergic and directly inhibit post-synaptic AgRP neurons. Consequently, silencing DMH^LepR^ neurons is predicted to relieve this inhibition and thereby increase AgRP neuron activity. In support of this possibility, we observed transient hyperphagia and positive energy balance following TeTx-mediated inhibition of DMH^LepR^ neurons, but we are not yet able to confirm that this results from disinhibition of AgRP neurons. Nonetheless, AgRP disinhibition seems likely to explain both this early outcome and the preservation of body weight during fasting, given that AgRP activation promotes lipogenesis while impairing lipolysis (see Cavalcante-de-Albuquerque et al., 2019). Taken together, our findings are consistent with AgRP neuron disinhibition as a predicted consequence of TeTx-mediated silencing of DMH^LepR^ neurons, but ascertaining the precise contribution of AgRP disinhibition to the phenotype we observed is an important goal for future studies.

10) With regard to food entrainment and anticipation under the experiments in Figure 4, the results provide intriguing evidence that DMH-LepR neurons play a role in adaptation to food availability within limited time windows under light:dark (diurnal) conditions. Whether this reflects changes in a circadian behavioral process and/or a circadian mechanism is uncertain-the adaptive response is sustained as the authors note even in SCN-ablated animals so the link to clock mechanisms has remained mysterious (this aspect can be included through revision of the text and conclusions rather than new experimentation).

In addition to the need for alternate feeding windows and/or TRF duration to more fully characterize the ability of DMH^LepR^-silenced mice to adapt to TRF (see response to point 6 above), we agree with the need for future studies to better clarify the precise contribution of DMH^LepR^ neurons to food-entrainment. For this reason, and due to the spatial constraints of the “Short Report” format, we elected to be conservative in the interpretation of our findings, but have added text to the revised manuscript highlighting the need for additional work (see the Results and Discussion).

11) When the mice with LepR deletion in the DMH undergo fasting-refeeding, they first exhibit significantly less weight loss (Figure 2B), and then compensate with less rebound feeding, which is not further blunted by leptin (as opposed to the anorexic response to leptin in fasted controls). This initial observation makes it hard to interpret whether the lack of a leptin-mediated effect in TeTx animals is simply due to the attenuated weight loss during fasting in these animals.

We acknowledge that the initial refeeding difference may obscure the ability of leptin to further suppress refeeding in DMH^LepR^-silenced animals. In the revised manuscript, we include additional discussion of this issue and potential mechanism to reflect the nuances alluded to by the reviewers (see the response to point 9 above).